# A Whole Range of Cattle—An Isotopic Perspective on Roman Animal Husbandry in Lower Austria and Burgenland (Austria)

**DOI:** 10.3390/ani14172624

**Published:** 2024-09-09

**Authors:** Günther Karl Kunst, Micha Horacek

**Affiliations:** 1Vienna Institute for Archaeological Science, Research Network Human Evolution and Archaeological Sciences, University of Vienna, Josef-Holaubek-Platz 2 (UZA II), 1090 Wien, Austria; 2Department of Lithospheric Research, University of Vienna, Josef-Holaubek-Platz 2 (UZA II), 1090 Wien, Austria; untertrias@gmail.com

**Keywords:** archaeozoology, Roman period, osteometry, stable isotope analysis, cattle husbandry, Roman Austria

## Abstract

**Simple Summary:**

Cattle remains from the Roman period often indicate both small and large individuals. This can be evidenced through the variability in bone measurements related to the stature of living animals. It is widely believed that these differences are too great to be related to the size pattern of cows and bulls from a single population, instead pointing to the presence of different types. The smaller one is usually conceived as autochthonous, while the larger one is interpreted as—originally—introduced Roman cattle. Apparently, for the first time in Central Europe, people would use two or more breeds of the same domestic species. To determine the background of this new production regime, we analysed four stable isotope ratios of bone collagen from small and large specimens from Roman sites, including urban, rural, civilian, military, and ritual. If the two types were raised differently, this should be visible by the isotope ratios providing information about nourishment and origin. The results produced no consistent differences between small and large cattle; rather, these were raised side by side. Apparently, Roman agriculture was complex enough to support various breeds simultaneously, but further research on intra-site variability is needed.

**Abstract:**

In this study, we try to combine traditional archaeozoological biometry, based on outer bone measurements, with stable isotope analyses of bone collagen. Right from the start of archaeozoological research in Central and Western Europe, the important size variability in Roman domestic cattle has puzzled scholars. According to an established view, these differences in bone size are attributed either to the simultaneous presence of different types or even breeds or to the result of crossbreeding of smaller, native, and larger Roman cattle. Likewise, the episodic import of large-sized animals has been considered. First, we selected thirty proximal phalanges of cattle from three sites including five archaeological contexts from eastern Austria (Roman provinces of Noricum and Pannonia). The bone sample comprised the whole hitherto observed metric variability in Roman provincial cattle, and we tried to include minimal and maximal specimens. The results from stable isotope analyses (δ^15^N, δ^13^C, δ^18^O, δ^2^H) carried out on thirty proximal phalanges indicated that isotope signals were rather site-specific and, generally, not related to bone size. Therefore, we conclude that at least in the area investigated, small and large cattle types were raised and herded in the same areas and not spatially separated. There are, however, uncertain indicators of intra-site differences in isotope signals related to bone size, which should be checked on much larger sample sets.

## 1. Introduction

Size trends in domestic cattle constitute a recurring topic in the zooarchaeology of the Roman period, starting from the early days when this field of research became established [1] up to overviews of the second half of the 20th c. [2,3,4]. Far from being a redundant or obsolete theme, the study of both phenotype and use of cattle in the Roman period remain relevant research goals today. This is shown by a recently growing number of academic papers on these topics, e.g., [5]. This holds especially true for Central and Western Europe and also for the southern part of the continent [6,7,8]. Along with the introduction of certain livestock species (domestic birds, cats, donkeys, and mules) [3], the adoption of new butchery methods [9], the bringing about of structured deposits resulting from these and other craft activities [10] like specialised Roman industrial bone dumps [11] (p. 4), the appearance of large-sized cattle, commonly dubbed Roman, Italic, or Mediterranean, constitutes a key marker for many Roman animal bone assemblages. This may be even more evident in the realm of what would become the Western and Central European provinces, outside of the Mediterranean area. It separates Roman-period animal bone remains clearly, or at least in most cases, both from assemblages of the pre-Roman Iron Age and from coeval sites outside of the Empire. Large cattle bones may indeed be more widespread and more directly evidenced than any other indicators of change, e.g., like those linked to farming systems and the use of secondary products, meat curing, etc. However, the concept of these uniform temporal and geographical clines or hiatuses may be an over-simplification or bias resulting from a Central European perspective. They may be archaeologically highly visible and pertinent in core areas like the Danubian limes and northern Gaul [10], where Roman occupation appears as a clear-cut marker horizon. In parts of Britain, Belgium, and the Netherlands, size patterns may vary considerably according to region and site type, with little traces of change until well after the integration into the Empire. Through meta-analyses based on many sites and samples, a much more detailed and complicated picture can be drawn [10,12,13].

On the other hand, the appearance of large-sized cattle remains within Iron Age materials represents a real topos and a prolific research theme [14,15]. These findings are usually interpreted as Roman, or, more generally, Mediterranean, imports and are often regarded as a kind of exotic luxury item. Long-term meta-analyses for what would later become the Western Empire, namely, Italy and parts of Spain, indeed show a progressive size increase in the remains of cattle already starting in the Iron Age [5,8]. Along with other innovations in livestock management, this probably resulted from raised connectivity in socio-economic systems, which antedates Roman occupation proper. These trends were therefore not primarily caused by integration into the Empire, if only certainly accelerated by it. Throughout the Empire, regional differences persisted, and meat diets changed to various degrees in different regions. Changes were not homogenous within proviniciae or neighbouring regions and did not affect all sites in the same way. Urban sites, villae, and military sites across the Empire displayed a higher degree of change, whereas rural sites more often retained forms of production that were closer to those of the Iron Age [5] (p. 409).

Returning to the Iron Age–Roman transition period in Central Europe, the changes appear to be more abrupt and are therefore not the result of long-term local development but rather of a sudden outside influence—the Roman occupation and settlement, commonly referred to as Romanization. Regarding the animals themselves, the size increase appears generally far too widespread and uniform to be explained by the presence of Roman draught oxen alone. Likewise, increased care and better nourishment are today no longer seen as sole factors that alone could account for the re-shaping of a local cattle population [16]. Thus, the import of a large-sized breeding stock, not only of single animals, is now generally assumed. Regarding the Rhine and Danube provinces, the traditional and most widely accepted narrative about how the observed size change came about is as follows: Local (native, autochthonous) breeding stock was crossbred with imported large-size cattle; together with improved management methods, this resulted in a general increase in the average size of the whole population [17]. Clearly, this affects both individual bone measurements and estimates of withers heights based on lengths of complete bones and, equally, statistical parameters calculated for both sets of data, especially if they are derived from larger samples. Like already stated in the quotations above, the onset of this process differs along with the region or province under study. According to Peters [4] (p. 57ff.), this occurred at the Magdalensberg and in Southern Germany right at the beginning of the Roman occupation, but only later in the Lower Rhine valley. One further question concerns the issue of whether native and local cattle stocks would finally constitute a single population with great variability [17] or if they continued to exist side by side, as two separate types, conceivably fulfilling different economic purposes. Here, regional differences and morphology come into play because the two (or more?) types are sometimes believed to be separable by the shapes of their horncores, by the micromorphology of bones, namely, of their articulations [18], and by enamel pattern on teeth [15]. Apparently, the small breed or type was maintained, at least north of the Alps, throughout the Roman period [4] (p. 58). Also, according to [13] (p. 485), small-sized cattle persisted throughout the Roman period in Belgium, albeit in low numbers. Lepetz [10] (pp. 38–39) provides a chronologic development for northern France: If the total of all samples from one period is considered, the frequencies of the large type occupied about 50% in the first c. AD, rising to over 90% in the third c. By the fifth c., the small form had almost completely disappeared.

Meanwhile, regional overviews and studies focused on certain sites have shown that local developments may be more complex than simply the gradual substitution of small native types by large Roman cattle. In the case of the Iron Age–Roman transition period in Raetia (southern Bavaria and adjacent areas), local rural residents specializing in cattle rearing apparently adopted the important stock earlier than the truly Romanized populations from nearby urban settlements [19]. Other studies found that increases in bone measurements during the Roman period were sometimes discontinuous [20] or ephemeral and characterized by reversals [16]. Overviews for northern Gaul [21], South-East Britain [12], and the Netherlands [11] discuss the strong influence of regional economic developments, Romanization and ethnicity, and settlement type on the respective representation of small- and large-sized cattle. They draw a more fine-grained picture across areas and through time, but they do not principally question the concept of an average size increase during the Roman era.

In this paper, we investigate stable isotope ratios from cattle bones of different size classes that can tentatively be attributed to the small and large types, respectively. Thereby, we focus on materials and contexts that exhibit wide metric variability, corresponding to the known Roman provincial range, where the archaeological background is well understood. The underlying hypotheses are as follows: if small- and large-type cattle were both reared, kept, and herded in about the same locations, their isotope ratios should agree. If, however, the two types originated from different production systems, including different regions, differences could become apparent. Of course, this still leaves the possibility for all kinds of transitory situations. Ideally, small and large types could be set apart in the same manner, if, say, native cattle were constantly imported from the interior of the Alps or from the Barbaricum north of the river Danube, while the Roman stock was reared locally. But this is just one of the many scenarios possible. Therefore, we consider our results as preliminary and experimental because still too little is known about the local picture of stable isotope ratios in cattle and our sample size is small. This study is meant as a starting point for further analysis. To our knowledge, this is the first study that deals with intra-species variability in stable isotope ratios other than strontium in Roman cattle and links it to size and—potentially—breed diversity. We are well aware of a series of recent studies on Roman cattle mobility based on Sr-isotopy, e.g., [22,23].

### 1.1. Assessing Size in the Skeletal Remains of Roman Cattle

Information on the phenotypic size of an archaeological cattle population can be obtained from osteometric measurements of specific skeletal elements and by evaluating these data across the various skeletal parts present in the sample. If measurements of long bone lengths are available, estimation of withers heights can be calculated. These may all be either awkward or impractical in the case of small or heavily fractured materials, where no single element is likely to produce enough measurements. A more elegant way is made possible by calculating logarithmic size indices (LSI values, the log size index method) of measurements taken from different elements. This method calculates the decimal log of the ratio between a measurement and a standard value [11] (p. 6); it allows for simultaneous consideration of many bone measurements from an analytical unit. However, while linear measurements of skeletal elements have been standardized for mammals and birds for some time [24], there is less uniformity in the application of the log size index method, that is, in the selection of measurements and how they are processed. In the fundamental paper on size development in cattle from northern Switzerland from the late La-Tène period to the early Medieval [17] (Breuer et al. 1999), all depth and breadth measurements of long bones and all measurements of short bones from the appendicular skeleton were used, excluding only the distal phalanges. Each measurement was included individually, and no averages per specimen were calculated. Groot and Albarella [11], for instance, suggest the following more subtle selection of measurements: they analysed length, width, and depth measurements separately and included values obtained from teeth. However, only one measurement per bone per anatomical plane was used. In a recent summary of the metrics of Roman cattle assemblages from Austria [25], we followed the criteria indicated by [17], calculating each bone measurement independently and excluding lengths of long bones and any metric data from the dentition.

However, the generation and processing of osteometric values is not the whole story. The delimitation of the analytical unit or contextual aggregation [26] from which these values are calculated may be of equal importance. Key factors here are the spatial and temporal resolution of the unit, which may comprise a single archaeological structure, a building, a whole settlement, or even a region or province. If you draw the limits too tight, you run the risk of receiving an insufficient database that may not be representative at all. On the other hand, if you amalgamate the whole animal bone material, e.g., of a complex Roman settlement, the samples may derive from large areas, from different contexts, and are also likely to cover a longer time span. Any regional, functional, and chronological trends might be completely obliterated. Especially in older studies, when contextual and stratigraphic information was limited, lumping was often the only method. In her pioneer dissertation on more than 12,000 cattle bones from the Roman site of Lauriacum (Upper Austria), Baas [27] discriminates only between the civil town and the military fort, providing additional information on certain contexts. Riedl’s study on Traismauer (Lower Austria) [28] comprises a rather large bone deposit from the first to fourth centuries A.D., with more than 8700 cattle remains. Still, these older studies have their merits as they convey the overall metric variability in Roman cattle remains from a defined region. In both papers, this large variability is seen as the outcome of the synchronous presence of native and Roman cattle. A comparative investigation by one of us [25] on cattle variability based on LSI values from ten contextual aggregations of the same area, central and eastern Austria, indicated that the maximal dispersion of measurements remained rather stable throughout the Roman period. However, the respective representation of small and large types may differ tremendously. Nowadays, most scholars compromise when defining and delimiting their aggregations, especially if meta-studies are envisaged [11] (pp. 3–4). Generally, more attention is paid to temporal resolution. An attempt is made to define analytical units of both a reasonable size and a reliable chronological position. An approximative overview of the correlation between sample size and the feasibility of different zooarchaeological analyses is provided by Marom and Bar-Oz [26]. The aspect of intra-specific osteometric comparisons is not included because it seems to be of interest only in certain archaeological situations, like the one discussed here.

### 1.2. Stable Isotope Ratios to Investigate Cattle Diet and Origin

Stable isotope ratios of carbon and nitrogen (δ^15^N and δ^13^C) in bone collagen reflect the δ^13^C and δ^15^N values of a diet, with a predictable offset (ca. 3–5‰ for δ^15^N; ca. 5–6‰ for δ^13^C) [29,30,31]. This enables reconstructions of what foods were consumed. For example, elevated δ^15^N values in herbivores can be linked to the consumption of fertilised crops [32], the consumption of plants growing on soils with larger nitrogen availability [33], or aridity [34]. Low δ^13^C values have been considered to indicate the consumption of forest resources [35], and/or in combination with feed grown at higher altitudes [36], high values indicate the consumption of feed (at least partially) consisting of C4 plants (e.g., [37,38]), or feed of plants that experienced significant drought stress [39,40], and references therein. Bone collagen stable isotope ratios of oxygen and hydrogen (δ^18^O and δ^2^H) mainly reflect the stable isotope ratio of consumed water, which is influenced by differences in temperature to a large extent and can therefore largely indicate whether an animal was introduced from an area with a different climate [38,39,41,42].

## 2. Materials and Methods

### 2.1. Material—Cattle Bones Used in This Study

Metric values and the results of stable isotope analysis are indicated in Table 1; the location of the study sites is provided in Figure 1. The principal idea of this study is to investigate the relationship between osteometric variability and stable isotope analysis on an intra- and inter-site scale. For this approach, it is important that osteometric variability is evaluated in accordance with the overall local Roman provincial picture. It also needs bone specimens that can be measured and allocated inside the supposed native-Roman continuum. Further, bones should derive from clearly defined contexts with known stratigraphic positions and high temporal resolution. Quickly sealed, single-phase archaeological features are preferred. These requirements cannot always be met, because Roman animal remains often accumulate within rubble and construction layers, usually with a multi-phase taphonomic history. For the following practical reasons, we decided to use proximal phalanges for isotope analyses: they are normally well represented and are early-fusing and sturdy—often being the only measurable bones within an assemblage with a complex taphonomic history. This is why they usually account for a greater part of LSI values within cattle samples [17]. In the mentioned osteometric paper [25], the measurements of proximal and middle phalanges contributed over 70% of the total dataset and, in certain sites, over 80%. There is, however, an innate problem with phalanges because they include four different skeletal elements each (anterior–posterior, medial–lateral). According to traditional knowledge and the literature [43], they differ in proportion, relative growth, and, consequently, in allometries. Therefore, they should be treated only with caution as a single entity. We believe this affects all sub-samples in the same way. In the present case, phalanges were not split up into anterior–posterior sub-groups. Their LSI values, regardless of whether from proximal or middle phalanges, were calculated and referred to the unspecified standard provided by the University of Basel [17]. Whether phalanges or the other elements were omitted had no sensible effect on the statistical parameters of the LSI distributions for a given sample [25]. That is, the LSI values obtained from phalanges can be considered representative of the cattle population. For isotopy, whenever possible, we tried to analyse specimens from the lower and upper end of the size spectrum. In theory, specimens from identical sites and size classes could also derive from the same cattle individuals, but this is unlikely because of the context data. If both native and Roman types are believed to be represented, native cows account for the smallest specimens and Roman bulls or oxen for the largest specimens. In all but one case, we relied on contextual aggregations already discussed by Kunst and Gál [25]. We nevertheless provide a short description of the relevant sites for the present study. If not indicated otherwise, data on cattle size are also taken from that paper. In small elements like phalanges, the size differences between small and large specimens may appear tremendous; if individual measurements are regarded, the largest can surpass the smallest by a factor of 1.5 or more.

From the vast settlement area of Carnuntum (Roman province of Pannonia; Lower Austria), we chose three pits from the sanctuary of Jupiter Heliopolitanus excavated around 1980. Details of the animal bone content and the cattle remains, in particular, are given in an earlier paper [44].

Pit G7 (formerly labelled C35) is rectangular with perpendicular walls, probably an abandoned shaft or well. It is situated near the eastern wall of the sanctuary. The homogenous infill contains terra sigillata from the time span AD 180–230; fitting sherds indicate a quick infill process. The cattle component is dominated by mandibles and metapodials and sticks out because of its size distribution: it presents the lowest average, median, and position of the IQR (interquartile range) of the ten analytical units compared in Kunst and Gál [25]. Obviously, mainly native cattle were processed here, producing a nearly unimodal LSI distribution [25] (p. 6). Nevertheless, the largest specimens also reach up to the maxima of the other aggregations. Of the five phalanges measured, three certainly belong to the small group and probably represent the native type; of the two remaining specimens, one corresponds to the lower part of the supposed Roman distribution, whereas the other competes with the largest phalanges of the whole dataset.

Pit G11 (formerly labelled L29) is a large, round pit in the southern part of the sanctuary, which produced an enormous, uniform faunal assemblage, probably representing the remains of one or several feasting events. The dominant cattle component indicates the processing of fresh carcasses at the spot. Though the excavation record shows a complex and multilayered infill, it seems to have become quickly consolidated around AD 180–220. The histogram of LSI values corresponds to a bimodal distribution and resembles the results from the nearby civil town, lacking only extremes at the upper and lower ends. Eight proximal phalanges were sampled, with four specimens attributed to the small and large groups each.

Pit M37 is a group of continuous pits, just outside of the wall, which only partly received debris from inside the sanctuary, containing rather normal urban refuse and partial skeletons of dogs and equids. It was included because of its strong cattle component. The distribution of LSI values is similar to pit G11 regarding median and average but with a more extended IQR. It, too, resembles the picture from various features of the civil town. Even though extreme LSI values are absent, we were able to select two quite large phalanges for isotope analysis. Two more were integrated into the large group, one was integrated into the intermediate Roman group, and two small specimens represented putative native stock. Among the contexts investigated here, M37 probably produced the metrically most evenly distributed sample.

At Halbturn (province of Pannonia, Burgenland), a rural cemetery, situated near a *villa rustica*, was excavated from 1988 until 2002. The animal bone material mainly derives from field ditches, which were filled up when this cemetery was enlarged [45]. This process can be roughly dated to the 2nd c. AD. According to an evaluation of the cattle measurements, mostly the upper half of the known Roman provincial variability is represented. This assemblage may indeed be devoid of native cattle, and the remaining variability in the four analysed phalanges can be explained by both sex groups of the large type. Thus, the lower limit of the Roman distribution can be estimated around a GLpe of 59 mm. Like in M37, because of its large dimensions, at least one specimen could be ascribed to a draught ox. Two more phalanges, deriving from unclear contexts underneath inhumation burials, do not correspond to the general preservation pattern. These may represent Iron Age relics from an earlier settlement. This hypothesis has been supported by radiocarbon dates indicating a younger Iron Age (330–200 BC) affiliation. These two bones, deriving from different features, were included here as the only non-Roman outgroup.

Finally, from Mautern-Favianis (province of Noricum, Lower Austria), situated about 100 km to the west of Carnuntum, four phalanges were included. These were collected from an abandoned well, overlain by late Roman stone masonry (site Melkerstrasse) [46]. A recent radiocarbon date puts this homogenous assemblage into the early 3rd c. AD (N. Kirchengast pers. comm.). It is totally dominated by cattle remains, with scooped long bones, scapulae, and ribs, indicating the curing of beef [47]. Further, it is the only material under discussion here not treated in [25], where, however, two aggregations from a nearby area of the vicus were discussed. While the earlier phases of Mautern-Favianis encompassed approximately the variability documented at Carnuntum, there is a shift towards higher values in the later periods. Although the metric variability in the cattle assemblage from the well has not been investigated so far, two of the four selected specimens occupy extreme positions on either end of the continuum, whereas the other two are fairly situated within the presumed lower part of the Roman distribution.

### 2.2. Osteometric Variability in the Cattle Sample

The osteometric variability in the proximal phalanges selected for isotope analysis is given in the scatterplot in Figure 2 based on the greatest peripheral length (GLpe) and proximal breadth (Bp), the combination of which provides a broad view of proportions. Much of the allometric variability is explained by shape differences between anterior and posterior phalanges, the latter being slenderer [43]. The density of the data points is biased because we tried to pick specimens with minimal and maximal values within each sample. The metric characters of the populations from the different contexts become apparent. While some of them (e.g., Mautern, M37) cover the whole range, Halbturn and G11 are restricted to the upper and lower areas, respectively. At Halbturn, only the possible prehistoric specimens are found within the lower part of the distribution. In G7, with its very low mean and median of LSI values, at least one specimen is found inside the very large group. Based on the LSI distribution and on literature data, we split the sample into three groups. According to Pucher [18], the overlap between small and large types in the measurement GLpe goes from about 53 mm to 60 mm. From personal observations and literature data [14], we know that Iron Age phalanges can reach values of up to 60 mm, as far as the normal population is concerned. Lepetz [10] indicates about 56 mm as the upper limit for the small type. For practical reasons, we split the distribution in Figure 1 into three groups, which were mainly determined by GLpe, not so much by Bp, as follows: small, with the upper limit of GLpe 56.4 mm, mainly defined by small specimens from G11; intermediate, with GLpe reaching from 59 to 62; and large, from GLpe 64,1 to the maximum. Large appears bipartite because of a discontinuity caused by Bp values. All samples are represented in each group, with the exception of Iron Age and Roman Halbturn, which are limited to the small and intermediate and large groups, respectively. In all likelihood, small should contain mainly or exclusively the native type, and intermediate and large should contain the Roman type. Conceivably, intermediate should indicate Roman cows, and large should indicate Roman bulls and oxen. There is some risk that specimens with a GLpe around 55 mm might represent small Roman cattle. The case numbers for small, intermediate, and large groups are 12, 8, and 10.

### 2.3. Stable Isotopes—Methods

All phalanges, which were brushed and cleaned in an ultrasonic bath before sampling, were sampled with a dental drill. Ca. 10 g aliquots were processed following the method described in [36,48] and the literature cited by the references mentioned. The extracted collagen was weighed into tin capsules and measured for their C- and N-stable isotope ratios by introducing them into a Thermo Flash EA elemental analyser (ThermoFisher, Bremen, Germany) and was weighed in silver capsules for the analysis of H- and O- isotopes by introducing them into a Thermo TC-EA (thermal combustion elemental analyser; ThermoFisher, Bremen, Germany). The evolving gas was flushed via a ConFlo IV (ThermoFisher, Bremen, Germany) into a delta V isotope ratio mass spectrometer (ThermoFisher, Bremen, Germany). The isotope ratio was given in the conventional δ notation in ‰ deviation with respect to internationally agreed standards (for carbon: Vienna PeeDee Belemnite (V-PDB), for nitrogen: N_Air_) and Vienna Standard Mean Ocean Water (V-SMOW) for H- and O-isotopes. Reproducibility was better than 0.3‰ for carbon and nitrogen isotopes, 0.7‰ for oxygen isotopes, and 5‰ for H-isotopes (1σ).

## 3. Results from Stable Isotope Analyses

In Figure 3 and Figure 4, we grouped the results from stable isotope analyses as bivariate scatterplots, according to the three size classes, by δ^15^N vs. δ^13^C and δ^18^O vs. δ^2^H. Neither plot reveals a consistent tendency of any size group regarding stable isotope composition. In Figure 3 (δ^15^N vs. δ^13^C), the extreme outer positions on all four sides are occupied by representatives of the small group, which exhibit an overall larger variability. Intermediate specimens are limited to the centre-right area of the distribution, with δ^13^C values over −21.5‰ and δ^15^N values between ca. 6‰ and 8.5‰. This reduced variability could either correspond to an artefact of the division criteria for the groupings or it could be caused by the influence of sites, where three locations, i.e., Mautern, Halbturn, and G11, are dominant within this size class. In Figure 4 (δ^18^O vs. δ^2^H), all size groups, including the intermediate specimens, appear to be distributed across the whole scatter. In the lower part of the diagram, with δ^18^O values less than 9‰, only large and intermediate specimens are represented. Small specimens are more frequent among higher δ^18^O values. The difference in δ^18^O values between small and large specimens is also the only one that proved statistically significant. However, it is mainly based on the good representation of Mautern and M37 in the lower part of the diagram. Regarding δ^2^H values, there is no discernible trend.

Figure 5 and Figure 6 contain the same data points as Figure 2 and Figure 3, but symbols for the five locations and the three size classes are added. In Figure 5 (δ^15^N versus δ^13^C; 29 data points), the four specimens from the Mautern well show the least variability of all: they occupy a small field on the lower centre of the distribution, with about the same value for δ^15^N and a variation of less than 1‰ in δ^13^C. The small phalanx, which is also the shortest specimen of the whole sample, exhibits the lowest δ^13^C value of the Mautern sample. Because all three size classes are represented, the values derive from at least three different individuals. The six specimens from Halbturn are found in the upper right corner of the scatter, considerably extending it towards higher δ^15^N values. The two Iron Age phalanges are found on the central and upper margins of this distribution. The Halbturn distribution appears to be the most detached and independent from the others. The seven data points for pit G11 form a loose scatter reaching from the centre to the left part of the distribution. In fact, they expand the distribution towards the left: three small and two large specimens exhibit the lowest δ^13^C values of the total. One intermediate specimen reaches far into the centre of the scatter. Within G11, the small phalanges tend to have lower δ^15^N values than the large ones. The five specimens from pit G7 follow on the right side and centre and, therefore, largely overlap with all other distributions. The only large individual exhibits the highest δ^13^C and δ^15^N values of these two pits from the sanctuary. Like in G11, small specimens exhibit the lowest δ^15^N values. The seven phalanges from M37 form a loose scatter going from the centre to the lower right margin of the distribution. Here, we observe the lowest values for δ^15^N and the highest for δ^13^C of the total, both represented by two small specimens. Even though there is a slight tendency for small specimens to exhibit lower δ^15^N values in G7 and G11, there is no general trend that sets apart the different size classes. However, some kind of grouping according to the site can be observed for all five entities.

This latter aspect also holds true, maybe to a lesser degree, for the δ^18^O vs. δ^2^H scatterplot, with 28 data points (Figure 6; one specimen from the Iron Age Halbturn produced no results). The four results for Mautern are all found in the lower left corner of the distribution. While their δ^18^O values are similar, they exhibit a considerable variation in δ^2^H values. The five data points for Halbturn show an important dispersion across the whole field, with two specimens on opposing ends. Except for the single Iron Age specimen marking the left lower corner, two large and one intermediate data points can be observed in the upper right centre, with one more intermediate outlier defining the upper right corner of the distribution. It is at least noteworthy that the Iron Age phalanx is detached from the Roman specimens. In pit G11, the dispersion is remarkable. The central unit is found in the upper left corner, with higher δ^18^O values. There is one small specimen to the right, with higher δ^2^H values, while a large specimen defines the lower left corner, with both low δ^2^H and δ^18^O values. In comparison, the field defined by pit G7 appears more compact. It roughly overlaps with the Roman distribution of Halbturn, with higher-than-average values for both δ^2^H and δ^18^O. Finally, the distribution for pit M37 is clearly bi-partite: the four large and the single intermediate specimens delimit the lower border of the total scatter, with δ^18^O values roughly between 8‰ and 9‰, varying δ^2^H values, and no overlap with other groups. The two small individuals are found on the upper right corner, in about the field occupied by G7 and Roman Halbturn. This is the only instance in this diagram where size classes from the same sample are clearly set apart, if only by δ^18^O values. Counter expectation, the overlap between G7 and G11, the two pits from the Carnuntum sanctuary, is only little in δ^13^C values; in δ^2^H values, there is definitely none.

## 4. Discussion

Clearly, any interpretations of the results presented in Figure 3, Figure 4, Figure 5 and Figure 6 are limited by the still too-small number of cases. Consequently, only very tentative statements are possible. Nevertheless, for the sites considered, the following becomes apparent: -There is no consistent relationship between modal size and stable isotope composition measured in Roman cattle bones;-The observed variability is explained by inter-site, or inter-context, differences rather than by size dependence.

The westernmost site, Mautern, and the easternmost site, Halbturn, are about 120 km apart from each other, and Carnuntum is situated less than 30 km to the north–northwest of Halbturn (Figure 1). It is therefore noteworthy that at least some regional signal is detectable for all five sites because the distributions are not randomly overlapping; there are clusters corresponding with sites and samples. Concerning temporal depth, all Roman materials can be ascribed to the second or third c. AD, with the two Iron Age bones from Halbturn as the only exception. G7, G11, and probably also Mautern well are from quickly filled-up structures and may represent close-knit aggregations. On the other hand, both the putative feasting assemblages from Carnuntum and the potential industrial infill from Mautern might contain remains from animals purchased from a wider territory—either for instant consumption and distribution or for the curing of meat. The bone assemblages from pit M37 and from the ditches at Halbturn probably took more time to accumulate. The collection from M37, but not necessarily the chosen specimens, may comprise the whole range of size groups and types present at the wider area of Carnuntum. In Roman Halbturn, it is most likely that the animals were kept at the local farm. It is also the only sample which does not comprise the regional size range of Roman-period cattle because the smaller class is missing. Within the total sample considered here, large-size draught oxen may be present in G7, M37, Halbturn, and Mautern (Figure 2). They do not show a common trend in either of the two stable isotope diagrams.

Regarding inter-site variability, the pits from Carnuntum and Mautern overlap strongly in the δ^15^N vs. δ^13^C diagram. Halbturn exhibits a more marginal position with both higher δ^13^C and, more importantly, δ^15^N values. This could be indicative of a generally warmer, more steppe-like environment or just different soil conditions at Halbturn, which is situated on the fertile gravel plain of Parndorf, still known as a breadbasket of the region. Further, around an affluent *villa rustica*, fertilised crops may have been easier available for the farm animals. Millet (*Panicum miliaceum*), a C4 plant, has indeed been found in archaeobotanical samples from Halbturn and is also known from the other Roman sites mentioned here [49]. As a fodder component, it may cause higher δ^13^C values. Animals may also have access to the salty marshes of the Lake Neusiedl area. Cattle raised at Mautern and Carnuntum, in contrast, could have profited from seasonally inundated wood pastures of the Danube floodplain.

In the δ^15^N vs. δ^13^C diagram in Figure 5, differences in the degree of patchiness among the samples are evident. The dense cluster for the data points from Mautern may be due to this cattle population forming a closed system and to the quick burial of their bones, as opposed to the more open, slowly accumulated assemblage from pit M37. The possible feasting pits G7 and G11 occupy an intermediate position in this case. In a recent study, it was shown that sheep living at the same time and feeding on identical resources resemble strongly in their δ^13^C and δ^15^N values, while those from different periods and grazing on different pastures differ [50]. Nothing for certain can yet be said about intra-site variability. In Figure 5, there might be a slight tendency for lower δ^15^N values in the small specimens from Carnuntum, but this is below statistical significance. If this trend was tested on a larger sample, it could correspond to the expected stereotype that cattle imported from the Barbaricum had less access to fertilised crops compared with their Roman congeners. As it stands now, the specimens from each of the sites and contexts were probably all raised locally. We would therefore definitely exclude that imported animals did play an important role, if not originating from regions with a similar isotopic pattern. If the hypothesis that the large Roman provincial metric variability includes both native (small) and Roman (large) cattle is correct, we may assume that both types, including the draught oxen, were reared and kept simultaneously in the same production systems and in about the same territory, or at least in areas that produced a similar isotope signal. Additional analyses (e.g., Sr isotopes) and enlarging the sample numbers will further substantiate this interpretation.

## 5. Conclusions

Measurements of cattle bones from Roman archaeological sites in Lower Austria in terms of their size in combination with stable isotope analyses showed no consistent pattern between isotope signatures and bone size. These results imply that small animals, interpreted as of non-Roman origin, and medium and large cattle, interpreted as of Roman origin, were reared in the same area and were probably of wider local origin, or they might have come from localities indistinguishable by stable isotope analysis. Future work will enlarge the number of investigated samples and include additional analytical methods.

## Figures and Tables

**Figure 1 animals-14-02624-f001:**
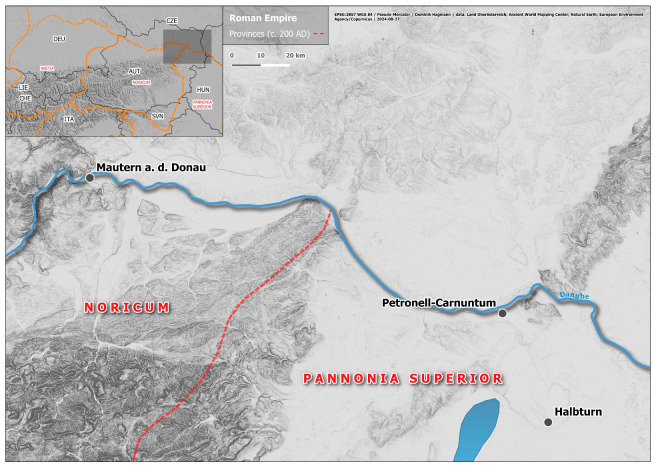
Topographic map with sites mentioned in the text and Roman provinces; source: EPSG:3857 WGS 84/Pseudo-Mercator|Dominik Hagmann|Data: Land Oberösterreich; Ancient World Mapping Center; Natural Earth; European Environment Agency/Copernicus|2 August 2024.

**Figure 2 animals-14-02624-f002:**
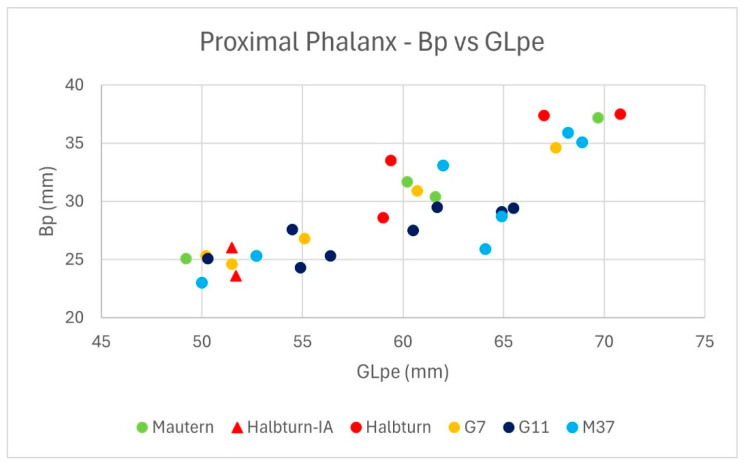
Bivariate scatterplot of proximal cattle phalanges used in this study, based on proximal breadth (Bp) and greatest peripheral length GLpe).

**Figure 3 animals-14-02624-f003:**
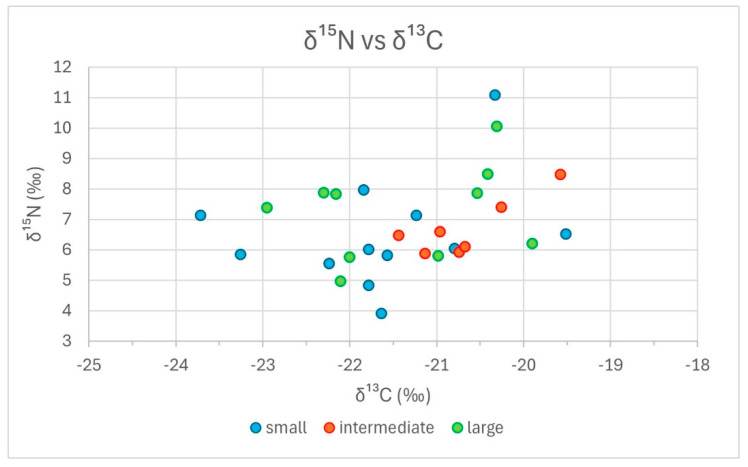
δ^15^N versus δ^13^C diagram; only size groups are indicated.

**Figure 4 animals-14-02624-f004:**
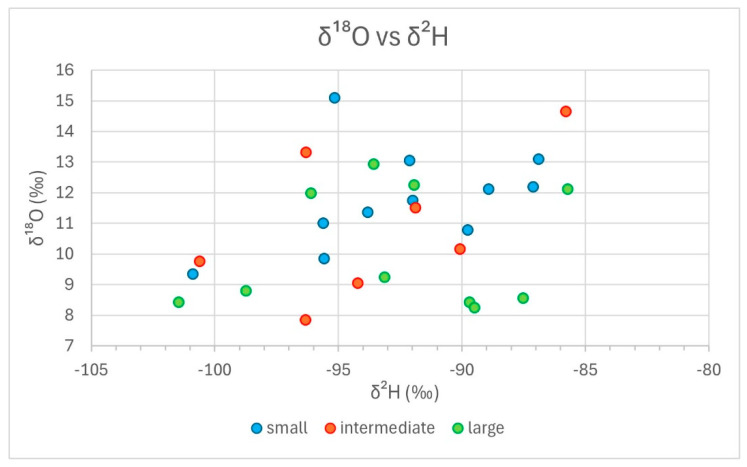
δ^18^O versus δ^2^H diagram; only size groups are indicated.

**Figure 5 animals-14-02624-f005:**
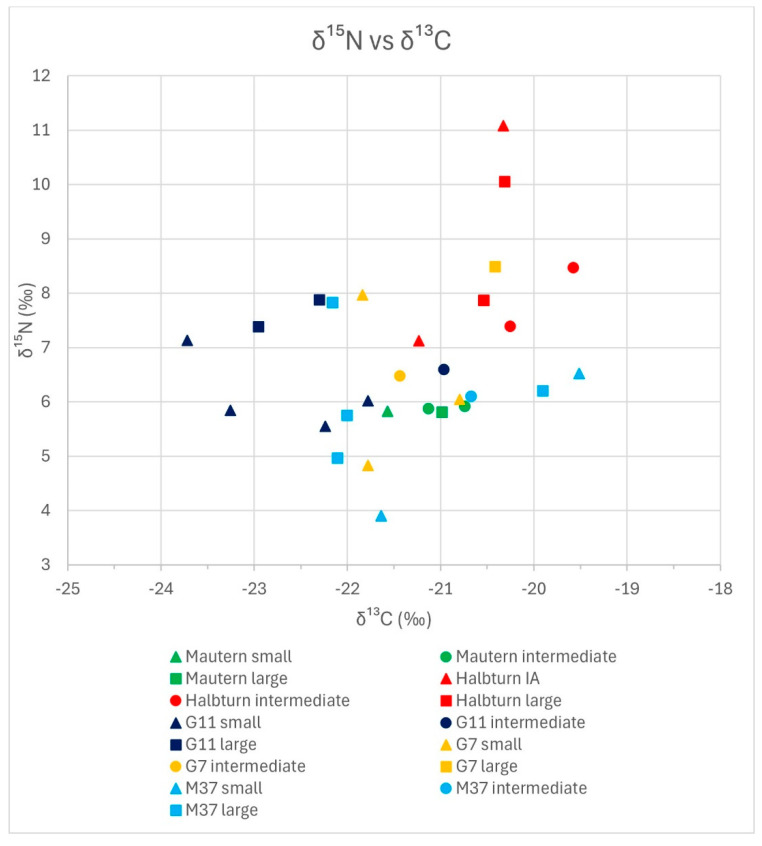
δ^15^N versus δ^13^C diagram, with contexts, sites, and size groups. IA: Iron Age.

**Figure 6 animals-14-02624-f006:**
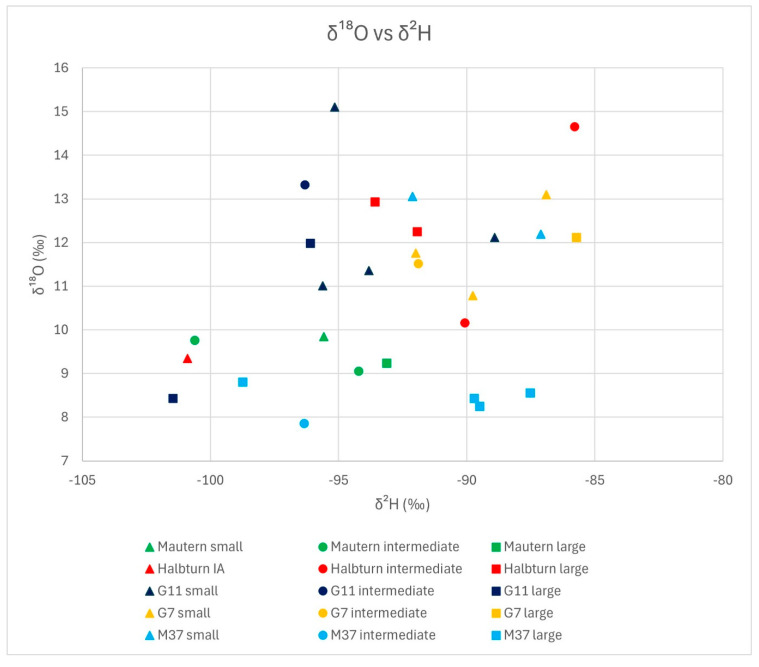
δ^18^O versus δ^2^H diagram, with contexts, sites, and size groups.

**Table 1 animals-14-02624-t001:** Metric values, attribution to size classes, and stable isotope ratios for the proximal cattle phalanges used in this study; laboratory numbers are only available for some of the specimens from Carnuntum. (Glpe: greater peripheral length, Bp: proximal breadth SD: smallest breadth of the diaphysis, Bd: distal breadth).

Lab.nr.	Site Context	Glpe (mm)	Bp (mm)	SD (mm)	Bd (mm)	Size Group	^13^C (‰)	^15^N (‰)	^2^H (‰)	^18^O (‰)
17-0071	Mautern	69.7	37.2	29.5	33.8	large	−21.0	5.8	−93.1	9.2
17-0072	Mautern	49.2	25.1	21.2	23.1	small	−21.6	5.8	−95.6	9.8
17-0073	Mautern	60.2	31.7	28.6	30.5	intermediate	−20.7	5.9	−94.2	9.1
17-0074	Mautern	61.6	30.4	24.7	28.2	intermediate	−21.1	5.9	−100.6	9.8
17-0075	Halbturn-IA	51.7	23.6	19.6	23.1	small	−20.3	11.1	−100.9	9.3
17-0076	Halbturn-IA	51.5	26			small	−21.2	7.1		
17-0077	Halbturn	59.4	33.5	26.8	31.7	intermediate	−19.6	8.5	−90.1	10.2
17-0078	Halbturn	59	28.6	21.9	28.3	intermediate	−20.3	7.4	−85.8	14.7
17-0079	Halbturn	70.8	37.5	30.9	35.6	large	−20.5	7.9	−91.9	12.2
17-0080	Halbturn	67	37.4	32.5	35.3	large	−20.3	10.1	−93.6	12.9
CA1	G11	54.5	27.6	23.5	27.7	small	−22.2	5.5	−93.8	11.4
CA2	G11	64.9	29.1	24.5	29.1	large	−23.0	7.4	−101.5	8.4
CA3	G7	67.6	34.6	29	34.8	large	−20.4	8.5	−85.7	12.1
CA4	G11	50.3	25.1	21.8	23.6	small	−21.8	6.0	−88.9	12.1
CA5	G11	61.7	29.5	27.5	30.5	intermediate	−22.1			
CA6	G11	65.5	29.4	23.7	28.2	large	−22.3	7.9	−96.1	12.0
CA7	G11	56.4	25.3	22.2	24.7	small	−23.3	5.8	−95.6	11.0
CA8	G11	60.5	27.5	22.1	26.6	intermediate	−21.0	6.6	−96.3	13.3
CA9	G11	54.9	24.3	21.2	22.7	small	−23.7	7.1	−95.1	15.1
CA10	G7	50.2	25.3	20.9	25.2	small	−21.8	8.0	−86.9	13.1
	G7	55.1	26.8	22	25.6	small	−21.8	4.8	−89.8	10.8
	G7	51.5	24.6	20.6	23.3	small	−20.8	6.0	−92.0	11.8
	G7	60.7	30.9	27.3	27.1	intermediate	−21.4	6.5	−91.9	11.5
	M37	50	23	19.2	22.4	small	−21.6	3.9	−92.1	13.1
	M37	52.7	25.3	22.9	24.6	small	−19.5	6.5	−87.1	12.2
	M37	62	33.1	31.2	33.2	intermediate	−20.7	6.1	−96.3	7.9
	M37	64.1	25.9	21.5	26	large	−19.9	6.2	−89.7	8.4
	M37	68.2	35.9	30.7	38.2	large	−22.0	5.8	−87.5	8.6
	M37	64.9	28.7	23.4	26.7	large	−22.2	7.8	−98.7	8.8
	M37	68.9	35.1	27.7	31.8	large	−22.1	5.0	−89.5	8.2

## Data Availability

Data are presented in the article.

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
