# Peer review of "A Whole Range of Cattle—An Isotopic Perspective on Roman Animal Husbandry in Lower Austria and Burgenland (Austria)"

_animals, 2024, doi:10.3390/ani14172624_

Round 1

Reviewer 1 Report

Comments and Suggestions for Authors

This paper is useful in that it indicates that the small local cattle and larger Roman cattle were, for the most part brought up  with each other in the same places.  There is little evidence that the Roman cattle were imported. It is unfortunately not easy to determine whether the residents were intentionally keeping different-sized cattle for different uses such as milking, draft, and plowing.  Variation in sizes of phalanges can be due to variation between fore- and hindlimbs and sexual dimorphism. See specifics below.

P. 5, Table 1: It would be very helpful for the authors to provide the terms that the abbreviations represent, i.e. Glpe, Bp, SD, and Bd.  These are known to those who are familiar with von den Driesch’s famous publication and who routinely do their own measurements, but many readers will find them mysterious.  On p. 7, lines 313 and 314, they do provide two interpretations: Glpe=greater peripheral length and Bp= proximal breadth. The reader is left to decipher the others, such as SD= smallest breadth of the diaphysis, not standard deviation, and Bd=distal breadth. Having the definitions in the Table 1 caption would be the most effective.

P. 8. Lines 315, 316: “the allometric variability is explained by shape differences between anterior and posterior phalanges, the latter being slenderer (Bartosziewicz 1993).”  This is a critical factor that can account for variations in sizes in the charts because some phalanges are fore and others are hind.  To the extent possible, the fore and hind should be dealt with separately or some mechanism should be used to take into account the differences between fore and hind proportions.  Otherwise, the hind phalanges of Roman cattle might overlap with the measurements of smaller local cattle fore phalanges.

 Lines 334-339: Now, we add another variable-sexual dimorphism.  So, the interpretations of the variation in measurements should consider whether the phalanx was fore or hind and male or female.  A slender hind phalanx of a male could overlap with a broader fore phalanx of a female.  If the authors cannot determine specifically hind vs fore or male vs female, then the variation in measurements can be difficult to sort out.

Lines 436-439: Interesting that there is some slight discernible difference between the two sites that are 120 km apart and some regional differences are apparent based on the isotopic results.

Line 450: Correction needed here: Halbturn, it is most likely the animals [insert WERE] kept at the local farm.

Line 482: Although earlier it was proposed that some of the cattle could have been introduced from a distance, they now conclude that is unlikely.  It is difficult to say with certainty, however, if the source had similar isotopic signals.  

Comments on the Quality of English Language

The English is overall acceptable, with only one notable error.  However, the overall composition does not make this a compelling or enjoyable article to read.  One grows weary of reading short sentences describing their results for a given site when there are so many, since it tends to jump from one site to another. Focusing on results, therefore, becomes a challenge.   I do not have a solution to this off-hand.

Reviewer 2 Report

Comments and Suggestions for Authors

Dear authors and editors,

The presented paper- "A hole range of cattle..an isotopic perspective...." is an interesting approach to a classical subject for Roman time archeozoology.

Based on the overall impression I have (as a zooracheologist) from this initial contact I had with the material, my main suggestion is to structure it as a "Case report/study" as the data is not completely supporting the scope of the paper.

The paper attempts to connect the osteometric data to some isotope analysis N/O/H and to try to establish a correlation based on a principle that these elements may be connected to different feeding habits, climates or different statuses of fodder in general.

The authors initiate their attempt by developing an ample introductory part on the effects of romanisation on the local stock and conclude on the main influence of cattle breeding and selection. What I find intriguing in this introductory part is the very ample section dedicated to this explanation, especially to approaching aspects from many works (but here I have a main suggestion- do not forget Bokony 1974- for sure you know about it) as the main source for Pannonia is there. On the other hand, given the extended section- why didn't you approach these general facts with situations from, say, Dalmatia? Noricum? Moesia? Dacia? Please take this as a suggestion, if this remains in this form- as a research paper (as a very similar situation is documented in these marginal Provinces, as well). And here I come with a suggestion for you- either you shrink a little this introductory part, limiting your elements to Pannonia and Noricum (id case study) or extend it to as many neighbouring Provinces as possible.

I would also like to ask for some more phrases and explanations in regards to isotope ratios and diet and origins (as the subchapter serves as a strong starting point for the initiative).

In subchapter "Osteometric variability...." please rearrange the figure 1....

Chapter 3 "Results..." is fine, it shows point-by-point the results and the explanations given by authors.

I strongly feel that there is a lot of data missing, as also authors state in their approach, as, in fact, data to extract info from is quite limited- 5 sites, and quite a limited number of samples from each site. I also feel that the initial selection of samples might be a biased source, (????). This is why I suggest again the structuring and presentation as "Case study" as the soundness of data is missing, having mainly hypotheses that are in fact not very well supported by obtained data.

Part 4- "Discussions" is not doing but to confirm the frail base we are all in (admitted from the first line by the authors)

References part I think is not meeting the MDPI/Animals demands in terms of bibliography- please correct. See also the referencing system within the text.

Once again, please o not take my observation as a personal one, I feel that the paper has merit, but still needs more data to support these initial statements made by authors. Thus, my suggestion is to change it into a Case study/Report and leave the conclusions open, less firm and allowing addition of more (substantial) data in the future.

Reviewer 3 Report

Comments and Suggestions for Authors

Kunst and Horacek paper deals with a hot topic in the research of the Roman period, namely the presence of different-sized Bos remains and the meaning of this variation. The authors intend to look at this problem by crossing traditional linear biometry of animal remains with stable isotope analysis. For that, a sample (n=30) of proximal phalanges from different sites (n=3) in eastern Austria was selected, considering their differences in size, aiming to comprise the biometrical variability, including smaller and larger specimens. These were then subjected to isotope analysis (δ15N, δ13C, 18 δ18O, δ2H). The authors conclude that the different cattle types by size were raised and herded in the same areas. An in-depth introduction to the topics being addressed demonstrates important knowledge in the discussion of the presence of different cattle sizes during the Roman period. The authors present their hypothesis to be tested while acknowledging the “preliminary and experimental” state of this research and the limitations of the sample size. This knowledge of the problem and caveats surrounding it is even clearer in the remaining sub-sections of the Introduction and the Materials and Methods, where several possible criticisms are already addressed by the authors (e.g., why use proximal phalanges while not separating by anterior/posterior and medial/lateral). The Materials and Methods are subdivided into different sub-sections where the selected cattle bones and their provenance are presented regarding contextual archaeological information and the linear osteometry variability. The stable isotope methods are also succinctly presented. The Results from the stable isotope analysis are described in a very complete manner and accompanied by the needed figures. Finally, the results are discussed and the main (preliminary) conclusions are presented. Overall this is a well-prepared manuscript where its “problems” are well acknowledged; at the same time, it is an interesting contribution that deserves publication and discussion among peers. Some minor issues should be addressed before publication:

·      Add a geographical location to the title

·      Add a map with the location of the sites used

·      Fig. 1 is not placed adequately in the text

·  Some recent publications could help improve the general introduction of the paper while adding information to the discussion/interpretations (e.g., https://doi.org/10.1016/j.quaint.2024.05.004; https://doi.org/10.5913/archbio04.05 and references therein)

·      Assessing size in skeletal remains of Roman cattle should be 1.1.

·      Stable isotope ratios to investigate cattle diet and origin should be 1.2.

·      Line 152. Change “3rd” for “distal” to be uniform with the rest of the article.

·      Table 1. Please mention the full acronyms of the measurements used in the caption. Also, separate Site and Context in Table 1 and mention that the measurements follow Driesch (1976).

·      Italicize terra sigillata (line 254), villa rustica (line 284, 465) and vicus (304)

·      When referring to deposits from pits and ditches instead of “fill” or “filling” use “infill” and “infilling” (e.g. lines 254, 255, 268)

·      Osteometric variability of the cattle sample should be 2.2.

·      Line 340- remove Micha from the subtitle

·      References do not follow the standards

Round 2

Reviewer 2 Report

Comments and Suggestions for Authors

Thanks for your notes and respomses.

At this stage I am happy with the comments and improvements to the text.